# Bayesian Inference of Temporal Task Specifications from Demonstrations

**Ankit Shah**
CSAIL, MIT
ajshah@mit.edu

**Pritish Kamath**
CSAIL, MIT
pritish@mit.edu

**Shen Li**
CSAIL, MIT
shenli@mit.edu

**Julie Shah**
CSAIL, MIT
julie_a_shah@mit.edu

## Abstract

When observing task demonstrations, human apprentices are able to identify whether a given task is executed correctly long before they gain expertise in actually performing that task. Prior research into learning from demonstrations (LfD) has failed to capture this notion of the acceptability of an execution; meanwhile, temporal logics provide a flexible language for expressing task specifications. Inspired by this, we present Bayesian specification inference, a probabilistic model for inferring task specification as a temporal logic formula. We incorporate methods from probabilistic programming to define our priors, along with a domain-independent likelihood function to enable sampling-based inference. We demonstrate the efficacy of our model for inferring specifications with over 90% similarity between the inferred specification and the ground truth, both within a synthetic domain and a real-world table setting task.

## 1 Introduction

Imagine showing a friend how to play your favorite quest-based video game. A mission within such a game might be composed of multiple sub-quests that must be completed in order to complete that level. In this scenario, it is likely that your friend would comprehend what needs to be done in order to complete the mission well before he or she was actually able to play the game effectively. While learning from demonstrations, human apprentices can identify whether a task is executed correctly well before gaining expertise in that task. Most current approaches to learning from demonstration frame this problem as one of learning a reward function or policy within a Markov decision process setting; however, user specification of acceptable behaviors through reward functions and policies remains an open problem [1]. Temporal logics have been used in prior research as a language for expressing desirable system behaviors, and can improve the interpretability of specifications if expressed as compositions of simpler templates (akin to those described by Dwyer et al. [2]). In this work, we propose a probabilistic model for inferring the temporal structure of a task as a linear temporal logic (LTL) specification.

A specification inferred from demonstrations is valuable in conjunction with synthesis algorithms for verifiable controllers ([3] and [4]), as a reward signal during reinforcement learning ([5], [6]), and as a system model for execution monitoring. In our work, we frame specification learning as a Bayesian inference problem.

The flexibility of LTL for specifying behaviors also represents a key challenge with regard to inference due to a large hypothesis space. We define prior and likelihood distributions over a smaller but relevant part of the LTL formulas, using templates based on work by Dwyer et al [2]. Ideas from universal probabilistic programming languages formalized by Freer et al [7] and Goodman et al [8], [9] are key to our modeling approach. Indeed, probabilistic programming languages enabled Ellis et al [10], [11] to perform inference over complex, recursively defined hypothesis spaces of graphics programs and pronunciation rules. We demonstrate the capability of our model to achieve greater

than 90% similarity between the ground truth specification and the inferred specification, both within a synthetic domain and a real-world task of setting a dinner table.

## 2 Linear Temporal Logic

Linear temporal logic (LTL), introduced by Pnueli [12], provides an expressive grammar for describing temporal behaviors. A LTL formula is composed of atomic propositions (discrete time sequences of Boolean literals) and both logical and temporal operators, and is interpreted over traces $[\boldsymbol{\alpha}]$ of the set of propositions $\boldsymbol{\alpha}$. The notation $[\boldsymbol{\alpha}], t \models \varphi$ indicates that $\varphi$ holds at time $t$. The trace $[\boldsymbol{\alpha}]$ satisfies $\varphi$ (denoted as $[\boldsymbol{\alpha}] \models \varphi$) iff $[\boldsymbol{\alpha}], 0 \models \varphi$. The minimal syntax of LTL can be described as follows:

$$\varphi ::= p \mid \neg\varphi_1 \mid \varphi_1 \vee \varphi_2 \mid \mathbf{X}\varphi_1 \mid \varphi_1\mathbf{U}\varphi_2 \tag{1}$$

$p$ is an atomic proposition; $\varphi_1$ and $\varphi_2$ are valid LTL formulas. The operator $\mathbf{X}$ is read as 'next' and $\mathbf{X}\varphi_1$ evaluates as true at time $t$ if $\varphi_1$ evaluates to true at $t + 1$. The operator $\mathbf{U}$ is read as 'until' and the formula $\varphi_1\mathbf{U}\varphi_2$ evaluates as true at a time $t_1$ if $\varphi_2$ evaluates as true at some time $t_2 > t_1$ and $\varphi_1$ evaluates as true for all time steps $t$ such that $t_1 \leq t \leq t_2$. In addition to the minimal syntax, we also use the additional first order logic operators $\wedge$ (and) and $\rightarrow$ (implies), as well as other higher-order temporal operators, $\mathbf{F}$ (eventually) and $\mathbf{G}$ (globally). $\mathbf{F}\varphi_1$ evaluates to true at $t_1$ if $\varphi_1$ evaluates as true for some $t \geq t_1$. $\mathbf{G}\varphi_1$ evaluates to true at $t_1$ if $\varphi_1$ evaluates as true for all $t \geq t_1$.

## 3 Bayesian Specification Inference

A large number of tasks comprised of multiple subtasks can be represented by a combination of three temporal behaviors among those defined by Dwyer et al [2] — namely, global satisfaction of a proposition, eventual completion of a subtask, and temporal ordering between subtasks. With $\varphi_{global}$, $\varphi_{eventual}$, and $\varphi_{order}$ representing LTL formulas for these behaviors, the task specification is written as follows:

$$\varphi = \varphi_{global} \wedge \varphi_{eventual} \wedge \varphi_{order} \tag{2}$$

We represent the task demonstrations as an observed sequence of state variables, $\boldsymbol{x}$. Let $\boldsymbol{\alpha} \in \{0,1\}^n$ represent a vector of a finite dimension formed by Boolean propositions. $\boldsymbol{\alpha} = f(\boldsymbol{x})$ (i.e., the propositions) are a function of the state variables of the system at a given time instant. The output of specification learning is a formula, $\varphi \in \boldsymbol{\varphi}$, that best explains the demonstrations, where $\boldsymbol{\varphi}$ is the set of all formulas satisfying the template described in Equation 2.

### 3.1 Formula Template

**Global satisfaction**: Let $\boldsymbol{\tau}$ be the set of candidate propositions to be globally satisfied, and let $\boldsymbol{T} \subseteq \boldsymbol{\tau}$ be the actual subset of propositions globally satisfied. The LTL formula that specifies this behavior is written as follows:

$$\varphi_{global} = \left(\bigwedge_{\tau \in \boldsymbol{T}} (\mathbf{G}(\tau))\right) \tag{3}$$

Such formulas are useful for specifying that some constraints must always be met — for example, a robot avoiding collisions during motion, or an aircraft avoiding no-fly zones.

**Eventual completion**: Let $\boldsymbol{\Omega}$ be the set of all candidate subtasks, and let $\boldsymbol{W_1} \subseteq \boldsymbol{\Omega}$ be the set of subtasks that must be completed if the conditions represented by $\pi_w; w \in \boldsymbol{W_1}$ are met. $\omega_w$ are propositions representing the completion of a subtask. The LTL formula that specifies this behavior is written as follows:

$$\varphi_{eventual} = \left(\bigwedge_{w \in \boldsymbol{W_1}} (\pi_w \rightarrow \mathbf{F}\omega_w)\right) \tag{4}$$

**Temporal ordering**: Every set of feasible ordering constraints over a set of subtasks is mapped to a directed acyclic graph (DAG) over nodes representing these subtasks. Each edge in the DAG

corresponds to a binary precedence constraint. Let $\boldsymbol{W_2}$ be the set of binary temporal orders defined by $\boldsymbol{W_2} = \{(w_1, w_2) : w_1 \in \boldsymbol{V}, w_2 \in \text{Descendants}(w_1)\}$, where $\boldsymbol{V}$ is the set of all nodes in the task graph. Thus, the ordering constraints include an enumeration of not just the edges in the task-graph, but all descendants of a given node. For subtasks $w_1$ and $w_2$, the ordering constraint is written as follows:

$$\varphi_{order} = \left( \bigwedge_{(w_1, w_2) \in \mathbf{W_2}} \left( \pi_{w_1} \rightarrow (\neg \omega_{w_2} \mathbf{U} \omega_{w_1}) \right) \right) \tag{5}$$

This formula states that if conditions for the execution of $w_1$ i.e. $\pi_{w_1}$ are satisfied, $w_2$ must not be completed until $w_1$ has been completed.

For the purposes of this paper, we assume that all required propositions $\boldsymbol{\alpha} = [\boldsymbol{T}, \boldsymbol{\pi}, \boldsymbol{\omega}]^T$ and labeling functions $f(\boldsymbol{x})$ are known, along with the sets $\boldsymbol{\tau}$ and $\boldsymbol{\Omega}$, and the mapping of the condition propositions $\pi_w$ to their subtasks. Under these assumptions, the problem of inferring the correct formula for a task is equivalent to identifying the correct subsets $\boldsymbol{T}$, $\boldsymbol{W_1}$, and $\boldsymbol{W_2}$, which explain the observed demonstrations well.

### 3.2 Specification Learning as Bayesian Inference

The Bayes theorem is fundamental to the problem of inference, and is stated as follows:

$$P(h \mid \mathbf{D}) = \frac{P(h)P(\mathbf{D} \mid h)}{\sum_{h \in \mathbf{H}} P(h)P(\mathbf{D} \mid h)} \tag{6}$$

$P(h)$ is the prior distribution over the hypothesis space, and $P(\mathbf{D} \mid h)$ is the likelihood of observing the data given a hypothesis. Our hypothesis space is defined by $\boldsymbol{H} = \boldsymbol{\varphi}$, where $\boldsymbol{\varphi}$ is the set of all formulas that can be generated by the production rule defined by the template in Equation 2. The observed data comprises the set of demonstrations provided to the system by expert demonstrators (note that we assume all these demonstrations are acceptable). $\boldsymbol{D}$ represents a set of sequences of the propositions, defined by $\boldsymbol{D} = \{[\boldsymbol{\alpha}]\}$.

#### 3.2.1 Prior specification

While sampling candidate formulas as per the template depicted in Equation 2, we treat the sub-formulas in Equations 3, 4, and 5 as independent to each other. As generating the actual formula, given the selected subsets, is deterministic, sampling $\varphi_{global}$ and $\varphi_{eventual}$ is equivalent to selecting a subset of a given finite universal set. Given a set $A$, we define $\texttt{SampleSubset}(A,p)$ as the process of applying a Bernoulli trial with success probability $p$ to each element of A and returning the subset of elements for which the trial was successful. Thus, sampling $\varphi_{global}$ and $\varphi_{eventual}$ is accomplished by performing $\texttt{SampleSubset}(\boldsymbol{\tau}, p_G)$ and $\texttt{SampleSubset}(\boldsymbol{\Omega}, p_E)$. Sampling $\varphi_{order}$ is equivalent to sampling a DAG, with the nodes of the graph representing subtasks. Based on domain knowledge, appropriately constraining the DAG topologies would result in better inference with fewer demonstrations. Here, we present three possible methods of sampling a DAG, with different restrictions on the graph topology.

---

**Algorithm 1** SampleSetsOfLinearChains

---

1: **function** SAMPLESETSOFLINEARCHAIN($\boldsymbol{\Omega}, p_{part}$)
2:      $i \leftarrow 1; \boldsymbol{C_i} \leftarrow []$
3:      $\boldsymbol{P} \leftarrow$ random permutation($\boldsymbol{\Omega}$)
4:      **for** $a \in \boldsymbol{P}$ **do**
5:          $\boldsymbol{C_i}$.append($a$)
6:          $k \leftarrow$ Bernoulli($p_{part}$)
7:          **if** $k = 1$ **then**
8:             $i = i + 1; \boldsymbol{C_i} \leftarrow []$
9:      **return** $\boldsymbol{C_j} \,\forall\, j$

---

**Linear chains:** A linear chain is a DAG such that all subtasks must occur in a single, unique sequence out of all permutations. Sampling a linear chain is equivalent to selecting a permutation from a uniform distribution and is achieved via the following probabilistic program: for a set of size $n$, sample $n - 1$ elements from that set without replacement, with uniform probability.

Table 1: Prior definitions and hyperparameters.

| Prior | $\varphi_{order}$ | Hyperparameters |
|---|---|---|
| Prior 1 | `RandomPermutation(`$\boldsymbol{\Omega}$`)` | $p_G, p_E$ |
| Prior 2 | `SampleSetsOfLinearChains(`$\boldsymbol{\Omega}, p_{part}$`)` | $p_G, p_E, p_{part}$ |
| Prior 3 | `SampleForestofSubTasks(`$\boldsymbol{\Omega}, N_{new}$`)` | $p_G, p_E, N_{new}$ |

**Sets of linear chains:** This graph topology includes graphs formed by a set of disjoint sub-graphs, each of which is either a linear chain or a solitary node. The execution of subtasks within a particular linear chain must be completed in the specified order; however, no temporal constraints exist between the chains. Algorithm 1 depicts a probabilistic program for constructing these sets of chains. In line 2, the first active linear chain is initialized as an empty sequence. In line 3, a random permutation of the nodes is produced. For each element $a \in \boldsymbol{P}$, line 5 adds the element to the last active chain. Lines 6 and 8 ensure that after each element, either a new active chain is initiated (with probability $p_{part}$) or the old active chain continues (with probability $1 - p_{part}$).

**Forest of sub-tasks:** This graph topology includes forests (i.e., sets of disjoint trees). A given node has no temporal constraints with respect to its siblings, but must precede all its descendants. Algorithm 2 depicts a probabilistic program for sampling a forest. Line 2 creates a random permutation of the subtasks, $\boldsymbol{P}$. Line 3 initializes an empty forest. In order to support a recursive sampling algorithm, the data structure representing forests is defined as an array of trees, $\mathcal{F}$. The $i^{th}$ tree has two attributes: a root node, $\mathcal{F}[i]$.root, and a 'descendant Forest', $\mathcal{F}[i]$.descendant, in which the root node of each tree is a child of the root node defined as the first attribute. The length of the forest, $\mathcal{F}$.length, is the number of trees included in that forest. The size of a tree, $\mathcal{F}[i]$.size, is the number of nodes within the tree (i.e., the root node and all of its descendants). For each subtask in the random permutation $\boldsymbol{P}$, line 5 inserts the given subtask into the forest as per the recursive function `InsertIntoForest` defined in lines 7 through 13. In line 8, an integer $i$ is sampled from a categorical distribution, with $\{1, 2, \ldots, \mathcal{F}.\text{length} + 1\}$ as the possible outcomes. The probability of each outcome is proportional to the size of the trees in the forest, while the probability of $\mathcal{F}$.length $+ 1$ being the outcome is proportional to $N_{new}$, a user-defined parameter. This sampling process is similar in spirit to the Chinese restaurant process [13]. If the outcome of the draw is $\mathcal{F}$.length $+ 1$, then a new tree with root node $a$ is created in line 10; otherwise, `InsertIntoForest` is called recursively to add $a$ to the forest $\mathcal{F}[i]$.descendants, as per line 12.

---

**Algorithm 2** SampleForestofSubtasks

1: **function** SAMPLEFORESTOFSUBTASKS($\boldsymbol{\Omega}, N_{new}$)
2:     $\boldsymbol{P} \leftarrow$ random permutation($\boldsymbol{\Omega}$)
3:     $\mathcal{F} \leftarrow []$
4:     **for** $a \in \boldsymbol{P}$ **do**
5:         $\mathcal{F}$ =InsertIntoForest($\mathcal{F}$,$a$)
6:     **return** $\mathcal{F}$
7: **function** INSERTINTOFOREST($\mathcal{F}, a$)
8:     $i \leftarrow$ Categorical($[\mathcal{F}[1]$.size, $\mathcal{F}[2]$.size, $\ldots, \mathcal{F}[\mathcal{F}.\text{length}]$.size, $N_{new}]$)
9:     **if** $i = \mathcal{F}$.length $+ 1$ **then**
10:         Create new tree $\mathcal{F}[\mathcal{F}.\text{length} + 1]$.root $= a$
11:     **else**
12:         $\mathcal{F}[i]$.descendants = InsertIntoForest($\mathcal{F}[i]$.descendants, $a$)
13:     **return** $\mathcal{F}$

---

Three prior distributions based on the four probabilistic programs are described in Table 1. In all the priors, $\varphi_{global}$ and $\varphi_{eventual}$ are sampled using `SampleSubset(`$\boldsymbol{\tau}, p_G$`)` and `SampleSubset(`$\boldsymbol{\Omega}, p_E$`)`, respectively.

### 3.2.2 Likelihood function

The likelihood distribution, $P(\{[\boldsymbol{\alpha}]\} \mid \varphi)$, is the probability of observing the trajectories within the data set given the candidate specification. It is reasonable to assume that the demonstrations are independent of each other; thus, the total likelihood can be factored as $P(\{[\boldsymbol{\alpha}]\} \mid \varphi) = \prod_{\{[\boldsymbol{\alpha}]\}} P([\boldsymbol{\alpha}] \mid \varphi)$.

The probability of observing a given trajectory demonstration is dependent upon the underlying dynamics of the domain and the characteristics of the agents producing the demonstrations. In the absence of this knowledge, our aim is to develop an informative, domain-independent proxy

for the true likelihood function based only on the properties of the candidate formula; we call this the 'Complexity-based' (CB) likelihood function. Our approach is founded upon the classical interpretation of probability championed by Laplace [14], which involves computing probabilities in terms of a set of equally likely outcomes. Let there be $N_{conj}$ conjunctive clauses in $\varphi$; there are then $2^{N_{conj}}$ possible outcomes in terms of the truth values of the conjunctive clauses. In the absence of any additional information, we assign equal probabilities to each of the potential outcomes. Then, according to the classical interpretation of probability, for candidate formula $\varphi_1$, defined by subsets $\boldsymbol{T_1}, \boldsymbol{W_1^1}$ and $\boldsymbol{W_2^1}$; and $\varphi_2$, defined by subsets $\boldsymbol{T_2}, \boldsymbol{W_1^2}$, and $\boldsymbol{W_2^2}$, the likelihood odds ratio is defined as follows:

$$\frac{P([\boldsymbol{\alpha}] \mid \varphi_1)}{P([\boldsymbol{\alpha}] \mid \varphi_2)} = \begin{cases} \frac{2^{N_{conj_1}}}{2^{N_{conj_2}}} = \frac{2^{|\boldsymbol{T_1}|+|\boldsymbol{W_1^1}|+|\boldsymbol{W_2^1}|}}{2^{|\boldsymbol{T_2}|+|\boldsymbol{W_1^2}|+|\boldsymbol{W_2^2}|}}, & [\boldsymbol{\alpha}] \models \varphi_2 \\ \frac{2^{N_{conj_1}}}{\epsilon} = \frac{2^{|\boldsymbol{T_1}|+|\boldsymbol{W_1^1}|+|\boldsymbol{W_2^1}|}}{\epsilon}, & [\boldsymbol{\alpha}] \nvDash \varphi_2 \end{cases} \qquad (7)$$

Here, a finite probability proportional to $\epsilon$ is assigned to a demonstration that does not satisfy the given candidate formula. With this likelihood distribution, a more restrictive formula with a low prior probability can gain favor over a simpler formula with higher prior probability given a large number of observations that would satisfy it. However, if the candidate formula is not the true specification, a larger set of demonstrations is more likely to include non-satisfying examples, thereby substantially decreasing the posterior probability of the candidate formula. The design of this likelihood function is inspired by the size principle described by Tenenbaum [15].

A second choice for a likelihood function, inspired by Shepard [16], is defined as the SIM model by Tenenbaum [15]. We call this the 'Complexity-independent' (CI) likelihood function, and it is defined as follows:

$$P([\boldsymbol{\alpha}] \mid \varphi) = \begin{cases} 1 - \epsilon, & \text{if } [\boldsymbol{\alpha}] \models \varphi \\ \epsilon, & \text{Otherwise} \end{cases} \qquad (8)$$

### 3.2.3   Inference

We implemented our probabilistic model in webppl [9], a Turing-complete probabilistic programming language. The hyperparameters, including those defined in Table 1 and $\epsilon$, were set as follows: $p_E, p_G = 0.8$; $p_{part} = 0.3$; $N_{new} = 5$; $\epsilon = 4 \times \log(2) \times (|\boldsymbol{\tau}| + |\boldsymbol{\Omega}| + 0.5|\boldsymbol{\Omega}|(|\boldsymbol{\Omega}| - 1))$. These values were held constant for all evaluation scenarios. The equation for $\epsilon$ was defined such that evidence of a single non-satisfying demonstration would negate the contribution of four satisfying demonstrations to the posterior probability. The posterior distribution of candidate formulas is constructed using webppl's Markov chain Monte Carlo (MCMC) sampling algorithm from 10,000 samples, with 100 samples used as burn-in. The posterior distribution is stored as a categorical distribution, with each possibility representing a unique formula. The maximum a posteriori (MAP) candidate represents the best estimate for the specification as per the model. The inference was run on a desktop with an Intel i7-7700 processor.

## 4   Evaluations

We evaluated the performance of our model within two different domains: a synthetic domain in which we could easily vary the complexity of the ground truth specifications, and a domain representing the real-world task of setting a dinner table — a task often incorporated into studies of learning from demonstration ([17]).

If the ground truth formula is defined using subsets $\boldsymbol{T^*}, \boldsymbol{W_1^*}$, and $\boldsymbol{W_2^*}$ (as per Equations 3, 4, and 5), and a candidate formula $\varphi$ is defined by subsets $\boldsymbol{T}, \boldsymbol{W_1}$, and $\boldsymbol{W_2}$, we define the degree of similarity using the Jaccard index [18] as follows:

$$L(\varphi) = \frac{|\{\boldsymbol{T^*} \cup \boldsymbol{W_1^*} \cup \boldsymbol{W_2^*}\} \cap \{\boldsymbol{T} \cup \boldsymbol{W_1} \cup \boldsymbol{W_2}\}|}{|\{\boldsymbol{T^*} \cup \boldsymbol{W_1^*} \cup \boldsymbol{W_2^*}\} \cup \{\boldsymbol{T} \cup \boldsymbol{W_1} \cup \boldsymbol{W_2}\}|} \qquad (9)$$

The maximum possible value of $L(\varphi)$ is one wherein both formulas are equivalent. One key benefit of our approach is that we compute a posterior distribution over candidate formulas; thus, we report the expected value of $\mathbb{E}(L(\varphi))$ as a measure of the deviation of the inferred distribution from the

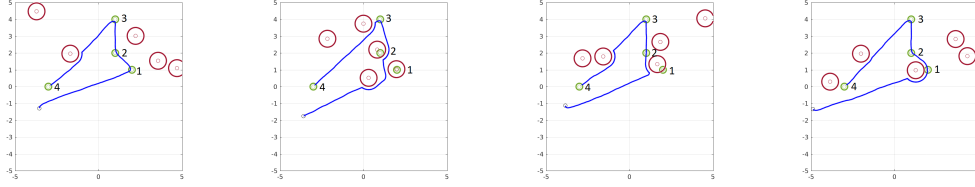

Figure 1: Example trajectories from Scenario 1. Green circles denote the POIs and the red circles denote the avoidance zones of threats.

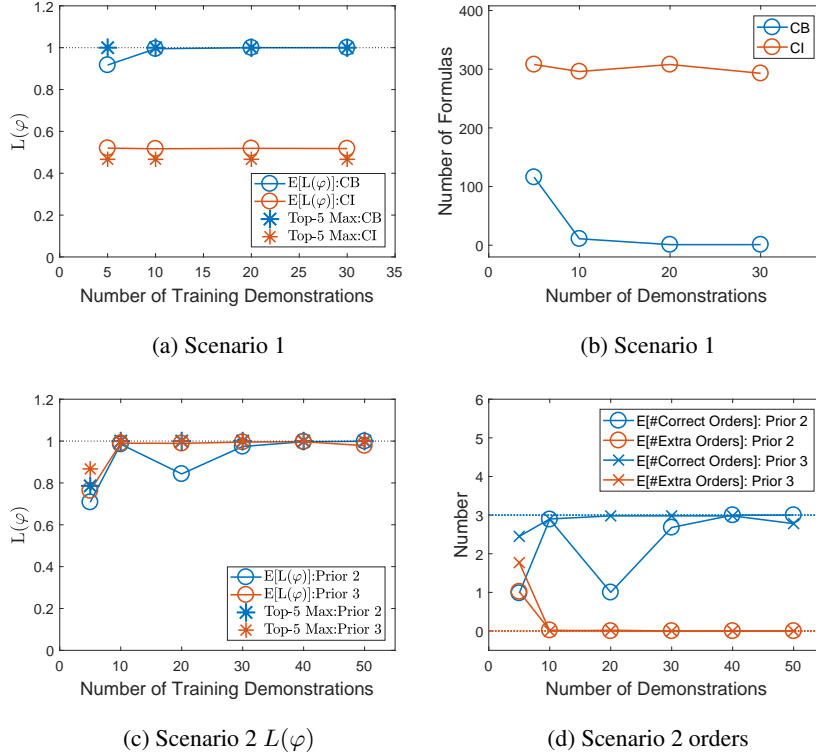

(a) Scenario 1

(b) Scenario 1

(c) Scenario 2 $L(\varphi)$

(d) Scenario 2 orders

Figure 2: Figure 2a depicts the results from Scenario 1, with the dotted line representing the maximum possible value of $L(\varphi)$. Figure 2b shows the number of unique formulas in the posterior distribution, Figure 2c indicates the $L(\varphi)$ values for Scenario 2, and Figure 2d depicts the correct and extra orderings inferred in Scenario 2. The dotted lines represent the number of orderings in the true specification.

ground truth. We also report the maximum value of $L(\varphi)$ among the top 5 candidates in the posterior distribution. We also classify the inferred orders in $W_2$ as correct if they are included in the ground truth, incorrect if they reverse any constraint within the ground truth, and "extra" otherwise. (Extra orders over-constrain the problem, but do not induce incorrect behaviors.)

We evaluated our approach against the temporal logic inference (TempLogIn) algorithm proposed by Kong et al [19]. TempLogIn mines parametric STL (PSTL) specifications, by conducting a breadth first search through a DAG induced by a partial ordering relation between PSTL formulas. Note that while our approach requires only positive examples, temporal logic inference must be trained on both positive and negative examples.

## 4.1 Synthetic Domain

In our synthetic domain, an agent navigates within a two-dimensional space that includes points of interest (POIs) to visit and threats to avoid. A predicate, $\omega_i$, is associated with each POI and evaluates as true if the agent is within a tolerance region of the given POI. Each threat has a predicate, $\tau_i$, associated with it, which evaluates as true if the agent enters an avoidance region for that threat.

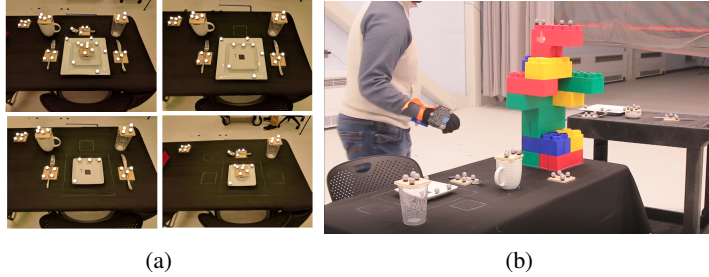

(a)                                                              (b)

Figure 3: Figure 3a depicts all the final configurations. Figure 3b depicts the demonstration setup.

Finally, propositions $\pi_i$ are associated with the accessibility of $i^{th}$ POI, and evaluate as true if the given POI is not within the avoidance region of any threat. The agent is programmed to visit the accessible POIs and avoid threats, as per the ground truth specification. In Scenario 1, we generated example trajectories in which the agent visits four POIs in a specific order $[1, 2, 3, 4]$. During each demonstration, five threat locations were sampled from a uniform distribution in the task space. Figure 1 depicts some of the demonstrated trajectories. In Scenario 2, we incorporated five POIs: $[1, 3, 5]$ must be visited in that specific order, while $\{2, 4\}$ can be visited in any order if accessible.

For Scenario 1, the posterior distribution was computed by using prior 1 (defined in Table 1) and both CB (Equation 7) and CI (Equation 8) likelihood functions for different training set sizes. The expected value and maximum value among the top 5 formula candidates of $L(\varphi)$ is depicted in Figure 2a.

We observed that the CB likelihood function performed better than the CI likelihood function at inferring the complete specification. Using the CI likelihood resulted in a higher posterior probability assigned to formulas with high prior probability that were satisfied by all demonstrations. (These tended to be simple, non-informative formulas; the CB likelihood function assigned higher probability mass to more complex formulas that explained the demonstrations correctly.) Figure 2b depicts the number of unique formulas in the posterior distributions. The CB likelihood function resulted in posteriors being more peaky, with fewer unique formulas as the training set size increased; this effect was not observed with the CI likelihood function.

For Scenario 2, the posterior distribution was computed using priors 2 and 3, as the ground truth specification did not lie in the support of prior 1. The expected value and maximum value among the top 5 formula candidates of $L(\varphi)$ are depicted in Figure 2c. Given a sufficient number of training examples, both priors were able to infer the complete formula; with 10 or more training examples, both priors returned the ground truth formula among the top 5 candidates with regard to posterior probabilities. Figure 2d depicts the correct and extra orders inferred in Scenario 2. Prior 3 assigns a larger prior probability to longer task chains compared with prior 2, but the two priors converge to the correct specification given enough training examples.

The runtime for MCMC inference is a function of the number of samples generated, the number of demonstrations in the training set, and the length of demonstrations. Scenarios 1 and 2 required an average runtime of 10 and 90 minutes for training set sizes of 5 and 50, respectively.

TempLogIn [19] required 33 minutes to terminate with three PSTL clauses. For both Scenarios 1 and 2, the mined formulas did not capture any of the temporal behaviors in Section 3.1, indicating that more PSTL clauses were required. With five and 10 PSTL clauses, the algorithm did not terminate within the 24 hours runtime cutoff. Scaling TempLogIn to larger formula lengths is difficult as the size of the search graph increases exponentially with number of PSTL clauses, and the algorithm must evaluate all formula candidates of length $n$ before candidates of length $n + 1$.

## 4.2 Dinner Table Domain

We also tested our model on a real-world task: setting a dinner table. The task featured eight dining set pieces that had to be organized on a table while avoiding contact with a centerpiece. Figure 3a depicts each of the final configurations of the dining set pieces. The pieces were varied in each configuration, but the position of a given piece on the table was constant across configurations, with positions marked on the table in order to guide the demonstrator. A predicate $\tau$ was associated with the centerpiece, encoding whether the demonstrators' wrists got too close to it. $\pi_i$ was associated

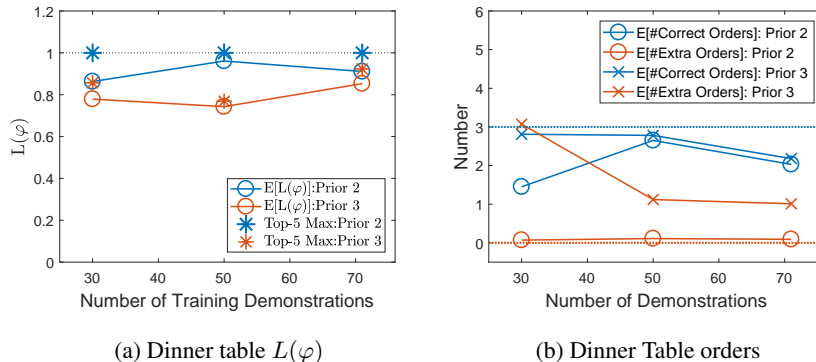

(a) Dinner table $L(\varphi)$            (b) Dinner Table orders

Figure 4: Figure 4a depicts the $L(\varphi)$ values for the dinner table domain, with the dotted line representing the maximum possible value. Figure 4b depicts the correct and extra orderings inferred within this domain. The dotted lines represent the number of orderings in the true specification.

with the $i^{th}$ dinner piece, and encoded whether that piece needed to be placed on the table. $\omega_i$ was associated with the $i^{th}$ dinner piece, and encoded whether it was at its correct and final position. In some of the configurations, the dinner plate, small plate and bowl were stacked on top of each other; in this case, the true specification would be to eventually position all the required dinner pieces by placing the dinner plate, small plate, and bowl, in that order. The state space $x$ consisted of the positions of each of the dinner pieces and the wrists of the demonstrator, all of which were tracked via a motion capture system. The truth values of $\omega_i$ and $\tau$ were evaluated using task-space region constraints defined by Berenson et al [20]. A total of 71 demonstrations were collected, and randomly sampled subsets of different sizes were used to learn the specifications. The expected value of $L(\varphi)$ and its maximum value among the top 5 candidates are depicted in Figure 4a; the number of correct and extra orders are depicted in Figure 1. With prior 2, our model correctly identified the ground truth in all cases. With prior 3, the inferred formula contained additional ordering constraints compared with the ground truth. Using all 71 demonstrations, the MAP candidate had one additional ordering constraint: that the fork be placed before the spoon. Upon review, this constraint was satisfied in all but four of the demonstrations.

## 5 Related Work

One common approach in prior research frames learning from demonstration as an inverse reinforcement learning (IRL) problem. Ng et al [21] and Abbeel et al [22] first formalized the problem of inverse reinforcement learning as one of optimization in order to identify the reward function that best explains observed demonstrations. Ziebart et al [23] introduced algorithms to compute optimal policy for imitation using the maximum entropy criterion. Konidaris et al [24] and Niekum et al [25] framed IRL in a semi-Markov setting, allowing for an implicit representation of the temporal structure of the task. Surveys by Argall et al [26] and Chernova et al [27] provided a comprehensive review of techniques built upon these works as applied to robotics. However, according to Arnold et al [1], one drawback of inverse reinforcement learning is that it is non-trivial to extract task specifications from a learned reward function or policy. Our method bridges this gap by directly learning the specifications for acceptable execution of the given task.

Temporal logics, introduced by Pnueli [12], are an expressive grammar used to describe the desirable temporal properties of task execution. Temporal logics have previously been used as a language for goal definitions in reinforcement learning algorithms ([5], [6]), reactive controller synthesis ([3], [4]), and domain independent planning [28].

Kasenberg and Scheutz [29] explored mining globally persistent specifications from optimal traces of a finite state Markov decision process (MDP). Jin et al [30] proposed algorithms for mining temporal specifications similar to rise time and setting time for closed-loop control systems. Works by Kong et al [31], [19], Yoo and Belta [32], and Lemieux et al [33] are most closely related to our own, as our work incorporates only the observed state variable (and not the actions of the demonstrators) as input to the model. Lemieux et al [33] introduced Texada, a general specification mining tool for software

logs. Texada outputs all possible instances of a particular formula template that are satisfied; however, it treats each time step as a string, with all unique strings within the log treated as unique propositions. Texada would treat a system with $n$ propositions as a system with $2^n$ distinct propositions — thus, interpreting a mined formula is non-trivial. Kong et al [31], [19] and Yoo and Belta [32] mined PSTL specifications for given demonstrations while simultaneously inferring signal propositions akin to our own user-defined atomic propositions by conducting breadth first search over a DAG formed by candidate formulas. Our prior specifications allow for better connectivity between different formulas, while using MCMC-based approximate inference allows for fixed runtimes.

We adopt a fully Bayesian approach to model the inference problem, enabling our model to maintain a posterior distribution over candidate formulas. This distribution provides a measure of confidence when predicting the acceptability of a new demonstration that the aforementioned approaches do not.

# 6 Conclusion

In conclusion we presented a probabilistic model to infer task specifications in terms of three behaviors encoded as LTL templates. We presented three prior distributions that allow for efficient sampling of candidate formulas as per the templates. We also presented a likelihood function that depends only on the number of conjunctive clauses in the candidate formula and is transferable across domains as it requires no information about the domain itself. Finally, we demonstrated that with our model inferred specifications with over 90% similarity to the ground truth, both within a synthetic domain and a real-world task of setting a dinner table.

# Acknowledgements

This research was funded in part by Lockheed Martin Corporation and the Air Force Research Laboratory. Approved for Public Release: distribution unlimited, 88ABW-2018-2502, 16 May 2018

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
