[Reviews · NeurIPS 2018]

Reviewer 1



The paper presents a method based on probabilistic programming for learning to infere temporal structures of tasks. The paper motivates the method with cases wherein humans can understand whether a task is properly executed or not, before having capacity to demonstrate it (which is the case in many problems). However, in the validation the approached problems do not fall in this description, so more complex problems related to one of the initial motivations of the paper would increase the value of the validation. The approach has been presented and validated for this concrete simple tasks with high level decisions. So, is it possible to extend or apply the method to tasks wherein the decisions are in lower level, like motor control for robot tasks? how would behave the learning method if the experiments are with problems that have stochastic transitions? The experiments show results of only one run, however in demonstrations there is always uncertainty and variance, different users would give different demonstrations in every trial. So it is important to conclude over several repetitions as it is typical in the literature. Human factors cannot be ignored in these methods. This approach is limited to problems wherein human teachers need to give high quality demonstrations to obtain good policies. How can be extended this approach in order to reach superhuman performance as the obtained by recent approaches of interactive learning? This discussion would need to aim to different applications with respect to the presented in the paper, since in those cases to provide high quality demonstrations is not a problem.

Reviewer 2



The authors introduce a probabilistic model for inferring task specification as a linear temporal logic (LTL) formula. This is encoded as three different behaviors, represented by LTL templates. The authors present linear chains, sets of LC and Forest of sub-tasks as prior distributions, as well as Complexity based and complexity independent domain-agnostic likelihood function. Given a set of demonstrations, the authors perform inference to obtain a posterior distribution over candidate formulas, which represent task specifications. The authors show that their method is able to recover accurate task specifications from demonstrations in both simulated domains and on a real-world dinner table domain. The authors provide a good background on LTL. The paper is well-written, and the explanations are clear. The main critique of this paper is very weak experiments. The setup of the experiments far too simple to accurately capture the value of this method. While the reader appreciates the use of a pedagogical toy example, it would be useful to do this on a more substantive dataset. Secondly, while the idea is neat, it is not necessarily novel. Similar approaches have been tried for both task structure learning and policy learning using the structure in hierarchical RL literature. See 3 example refs below of the many there can be. Also, there is little evidence that the current formulation enables scaling to complexities in the task that form the motivation of this work such as quest based games or multi-stage sequential tasks in robotics. Furthermore, the approach has problems in scaling due to the modeling choice. The framework needs to check all candidates of a given complexity (length n) before moving to higher orders. Additional comments: 1. "Ideas from universal probabilistic programming languages formalized by Freer et al [7] and Goodman et al [8], [9] are key to our modeling approach" Would be useful to say how. Additional refs to consider: - Sener, F., & Yao, A. (2018). Unsupervised Learning and Segmentation of Complex Activities from Video. arXiv preprint arXiv:1803.09490. -- can handle complex task graph structures. - Sohn, Sungryull, Junhyuk Oh, and Honglak Lee. "Neural Task Graph Execution." (2018). -- graph inference with constraints. - Krishnan, Sanjay, et al. "Transition state clustering: Unsupervised surgical trajectory segmentation for robot learning." Robotics Research. Springer, Cham, 2018. 91-110. -- linear chain inference as a clustering problem. Similar work has been performed for videos and such as well.

Reviewer 3



Summary: The authors extend Bayesian concept learning to infer Linear Temporal Logic (LTL) specifications from demonstrations. The concept space is that of a finite subset of all possible well formed formulas of LTL constrained to a conjunction between three well founded structures for capturing tasks (Global satisfaction, Eventual completion, Temporal ordering). The authors propose three different priors on the structure of the Temporal ordering (Linear chain, Set of linear chains, Forest of sub-tasks) which allow for efficient sampling of candidate formulas. The posteriors using the different priors have different explanatory power of the observed data. The explanatory power depends on how well the observed task adheres to the structure of the respective prior, which further provides insights on which temporal order structure structure that is more suitable for a specific demonstrated task. The authors propose a likelihood (likelihood odds ratio) which they evaluate together with the priors on both toy data and a real world task scenario with good results. Strengths: I really like the interaction between Bayesian machine learning and temporal logics in the paper. The paper reminds me of the first part of Chapter 3 about Bayesian concept learning in [1] where they show the elegance and power of Bayesian machine learning for learning from only positive demonstrations, although under much simpler circumstances. This is a direction I would like to see more of within the ML community. This paper is original to the best of my knowledge in the sense that the concept space is a finite subset of well formed formulas of a Modal logic (LTL) consisting of three proposed, for the problem well founded, structures as well as the proposal of three distinctly different but suitable priors used for the inference which are also empirically investigated. I deem the contributions significant and novel. The contributions are important for advancing this line of research into both the integration of ML and logics as well as opening up for future valuable contributions to the reinforcement and learning from demonstration communities. The paper is clearly written and have a fair presentation of both the probabilistic aspects of Bayesian machine learning and of Linear Temporal Logic (LTL). Everything is not easy to understand at first, but it is rather due to the many important technical details across the applied Bayesian inference and the temporal logic than anything else. The approach makes sense, is reasonable and the experimentation is suitable. The Algorithm listings help a lot and increase clarity of the paper. The paper seem to be technically correct and the quality of the paper is in my opinion high. The experiments are explained in detail and seem to be reproducible with access to the data. Weaknesses: For an audience which is less familiar with Bayesian machine learning, such as from other parts of the ML field or from the fields of more logic based AI, some parts of the paper can be made clearer. One example is the likelihood odds ratio in equation (7) which is not explicitly shown how it is used as a likelihood (e.g. in the relation to Bayes theorem). I understand why it is sufficient to state it as is, with the context of Bayesian model selection in mind, but it might not be self-evident for a wider audience. Other comments: It is my understanding that the three priors can be made into a single prior (e.g. with a probability of 1/3 to draw a sample from either of them). What would be the reason for doing this versus how you do it in the paper where you keep them separate as three different inferences? Some figures are very small. Especially Figure 1, but also Figure 2 and Figure 4. Figure 1 can be made bigger by limiting the plotting to y >= -2. The caption of Figure 2 seem to be inconsistent (e.g. "Scenario 1" and "Scenario 2 L(\phi)") [1] Murphy, K. P., Machine Learning - A probabilistic Perspective, 2012, The MIT Press. --- Thank you for your detailed author feedback.